# Three-Dimensional Quantitative Structure–Activity Relationship Study of Transient Receptor Potential Vanilloid 1 Channel Antagonists Reveals Potential for Drug Design Purposes

**DOI:** 10.3390/ijms25147951

**Published:** 2024-07-21

**Authors:** Beatrice Gianibbi, Anna Visibelli, Giacomo Spinsanti, Ottavia Spiga

**Affiliations:** 1Department of Biotechnology, Chemistry and Pharmacy, University of Siena, 53100 Siena, Italy; beatrice.gianibbi@student.unisi.it (B.G.); anna.visibelli2@unisi.it (A.V.); giacomo.spinsanti@unisi.it (G.S.); 2Centro della Scienza e della Tecnica, Polo Universitario Grossetano, Via Ginori 41, 58100 Grosseto, Italy; 3Competence Center ARTES 4.0, 53100 Siena, Italy

**Keywords:** TRPV1, artificial intelligence, 3D-QSAR, pharmacophore modeling

## Abstract

Transient receptor potential vanilloid 1 (TRPV1) was reported to be a putative target for recovery from chronic pain, producing analgesic effects after its inhibition. A series of drug candidates were previously developed, without the ability to ameliorate the therapeutic outcome. Starting from previously designed compounds, derived from the hybridization of antagonist SB-705498 and partial agonist MDR-652, we performed a virtual screening on a pharmacophore model built by exploiting the Cryo-EM 3D structure of a nanomolar antagonist in complex with the human TRPV1 channel. The pharmacophore model was described by three pharmacophoric features, taking advantage of both the bioactive pose of the antagonist and the receptor exclusion spheres. The results of the screening were implemented inside a 3D-QSAR model, correlating with the negative decadic logarithm of the inhibition rate of the ligands. After the validation of the obtained 3D-QSAR model, we designed a new series of compounds by introducing key modifications on the original scaffold. Again, we determined the compounds’ binding poses after alignment to the pharmacophoric model, and we predicted their inhibition rates with the validated 3D-QSAR model. The obtained values resulted in being even more promising than parent compounds, demonstrating that ongoing research still leaves much room for improvement.

## 1. Introduction

Chronic pain is a condition that causes subtle suffering, which seriously impacts the quality of life of individuals with neuropathic pain, cancerogenic diseases, and musculoskeletal injuries [1,2,3]. At present, several treatments are available, such as opioids and anti-inflammatory non-steroidal drugs, according to the patient’s needs and perceived pain intensity [4]. In the most serious cases, effective and potent opioids lack an optimal pharmacodynamics profile, triggering dependence and tolerance phenomena that limit drug usage over time or lead to abuse or addiction [5]. For these reasons, new therapeutic options would be advisable for a better resolution of such pathological conditions. Recently, transient receptor potential vanilloid 1 (TRPV1) was identified as a new putative target for the inhibition of pain transmission by direct action within the central nervous system (CNS) [6,7]. TRPV1 belongs to a large family of non-selective calcium-permeant cation channels, and it is mostly expressed in nociceptive neurons in the CNS, but it is also widely distributed in peripheral tissues, including vascular arteries, the liver, endometrium and ovaries, and skin [8]. TRPV1 activation triggers membrane depolarization and an increase in intracellular calcium levels, promoting a cascade of events with the involvement of peripheral nociceptors. After this rapid firing phase, the calcium ion intake can render such nociceptors insensitive to further stimuli, leading to a desensitization phase, responsible for a transient analgesic effect. Several factors contribute to the activation of the TRPV1 channel or later desensitization, such as ion concentration, pH, temperature, and small ligands [7]. Capsaicin falls into the latter category, together with a series of natural compounds able to act as TRPV1 agonists, which first activate the channel, but ultimately lead to desensitization and inhibition [9]. A variety of capsaicin derivatives have been designed to obtain the desired pharmacological properties [10,11], but at present, no synthetic or semisynthetic drug has been released to the market. Even widely used capsazepine is only used as a biochemical tool to study TRPV ion channels [12]. One of the major reasons for these limitations is linked to the onset of many adverse reactions as a consequence of low target selectivity and poor pharmacokinetics profiles. On the other hand, the discovery of TRPV1 antagonists has taken a few steps forward by introducing some competitive and non-competitive ligands as drug candidates. One of the competitive antagonists, SB-705498, was reported to be capable of penetrating the blood–brain barrier (BBB) [13]. Although in clinical trials it did not provide substantial improvement compared to the placebo, it never determined the onset of detrimental adverse reactions, such as hyperthermia, which is frequently associated with TRPV1 antagonism [14]. Starting from these encouraging results, the bioactive pyrrolidinyl-urea core was hybridized with the phenyl-thiazole moiety of partial agonist MDR-652 [15], obtaining several new synthetic derivatives [16]. These newly designed molecules were screened for TRPV1 inhibition by assessing their efficacy in abolishing the capsaicin-activated receptor’s activity at 10 µM. Their values ranged from 5 to 89%, where the maximum value reached overlapped with the one reported for reference antagonist SB-705498.

Most recently, a nanomolar allosteric antagonist of the TRPV1 channel was reported, namely SB-366791, which was capable of completely abolishing TRPV1 activity at 10 µM [17]. At the same time, the three-dimensional structure of the human TRPV1 channel in complex with SB-366791 was determined through cryo-electron microscopy (Cryo-EM) techniques at high resolution (2.3 Å) [18]. This result paved the way for the development of a rational design strategy to obtain even more effective antagonists of the channel, which had not been possible beforehand. Although the 2D similarity with SB-366791 is low compared to the ligands from Qiao et al., we investigated whether the same binding site could be shared by the hybridized series of compounds. In addition, the assessment of the efficacy of the newly designed compounds was made possible with high accuracy by the generation of three-dimensional descriptors within a quantitative structure–activity relationship model (3D-QSAR). This advanced modeling approach allows researchers to capture the intricate spatial and electronic features of molecules, providing a detailed understanding of how structural variations influence biological activity. By integrating these three-dimensional descriptors, 3D-QSAR models can more accurately predict the potency and efficacy of new compounds, facilitating the identification of promising candidates for further development. QSAR studies typically correlate observed activity with the structural properties of putative binders by exploiting mathematical models. By identifying key structural features that contribute to the activity, QSAR models enable the rational design of new compounds with enhanced properties. While such studies are usually performed on IC50 values as the target variable, we herein demonstrate that such an approach is exploitable for inhibition rates as well. Several methods for feature and descriptor generation have been developed so far, based on bi-dimensional and tri-dimensional fingerprints [19,20], whose combination led to the most informative results [21]. In earlier research, more advanced descriptors such as 4D, 5D, and even 6D have been proposed [22,23,24], although their full potential has yet to be fully realized in current research practices. Subsequently, the problem of feature selection has been addressed as well, with several dimensionality reduction algorithms being used to extract only the descriptors capable of capturing the key aspects of the training data. Principal component analysis (PCA) and uniform manifold approximation and projection (UMAP) are among the prominent methods employed for this purpose. PCA emphasizes linear relationships among data points, while UMAP excels in capturing non-linear relationships [25]. After the process of feature selection is completed, a mathematical model is employed to establish a correlation between the data and the target variable. A plethora of instances are available to researchers, including random forest (RF), support vector machine (SVM), Lasso regression, gradient boosting machine (GBM), k-nearest neighbors (k-NN), and neural networks (NN). Each of these methods offers distinct advantages, and combining them into an ensemble usually leads to greater chances of obtaining successful results [26]. As the performance of the model is strictly related to the quality of the input data, a useful strategy for feature generation is the Comparative Molecular Field Analysis (CoMFA), capable of integrating 3D structural information of molecules into predicted models, by generating grid-based molecular interaction fields (MIFs) [27]. Although the final result of applying this model is generally successful, a significant drawback is that such algorithms often depend on proprietary software, which is not always accessible to academic researchers.

This approach aims to illustrate that high performance can be achieved without the necessity for expensive, proprietary resources. By relying solely on freely available tools, we demonstrate that significant and reliable results are achievable even with limited resources.

## 2. Results

Starting from the hybridized synthetic compounds from Qiao et al., we generated up to 100 conformations each, to be screened for fitting on the three-dimensional Cryo-EM structure of the TRPV1 channel (PDB: 8GFA) [18]. We hypothesized that such compounds would be able to bind to the same allosteric pocket reported for the co-crystallized antagonist SB-366791, thus triggering similar pharmacodynamic effects. We selected three pharmacophoric features (two aromatic groups and one hydrophobic group) on the SB-366791 scaffold (Figure 1A,B).

Interestingly, both SB-705498 (used as reference) and the hybridized compounds demonstrated the ability to occupy the selected binding pocket while respecting all three pharmacophoric features, either involving one or multiple conformers, depending on the steric clashes with the receptor sidechains and the reciprocal orientation of the aromatic moieties. Some of the most active compounds of the set demonstrated extremely low root mean square deviations (RMSDs) compared to the SB-366791 bioactive pose. For instance, **3i** was reported to inhibit capsaicin-activated human TRPV1 channel activity of 83% and showed an RMSD of 0.079 Å. Overall, the deviation of the coordinates ranged from 0.030 to 0.876 Å, which was a surprising result given the variability across the 83 investigated molecules.

Such encouraging findings led to the development of a CoMFA-based 3D-QSAR relationship model, based on the three-dimensional coordinates of the antagonists within the evaluated binding site. The inhibition rate values were transformed into their negative decadic logarithm and combined with the screened structures for the 3D-QSAR model generation. In the 3D-QSAR study, MIFs were generated and analyzed for correlations with activity using partial least squares (PLS) regression and the variable selection methods implemented in Open3DQSAR software [28]. To assess the 3D-QSAR models’ performances, we divided the compounds into 20 different training and validation sets using different splitting schemes, always maintaining 90% of the data for training and 10% for validation. From these runs, we obtained a coefficient of determination (*R*^2^) of 0.78 with a standard deviation (*SD*) of 0.12, indicating consistent predictive accuracy across the different splits. The results obtained for the best model in terms of *R*^2^ are shown in Figure 2. The plot showcases how closely the predictions align with experimental data, highlighting the efficacy of the PLS regression model in both the training and validation phases.

After the validation of the 3D-QSAR model, we investigated whether we could design potentially more effective antagonists of the human TRPV1 channel. Following the rationale of most active ligands, we inserted key point modifications on the original scaffolds by combining the most remarkable moieties, while accounting for shape complementarity with the receptor pocket. The newly designed molecules that have been reported as hit compounds after pharmacophore screening are reported in Table 1. The best-fitting conformers of the hit compounds were analyzed with the newly generated 3D-QSAR model to predict their inhibition rates. The results spanned across a range of 25%-96% (Figure 3).

The most interesting compound, **11b**, showed a predicted value of 96%, being even more effective than previously reported antagonists. Intriguingly, it demonstrated the lowest RMSD across the series, with a value of 0.249 Å compared to SB-366791’s experimentally determined pose. Even though compound **11b** was almost lying outside our distribution of results, exceeding the interval of values |µ ± 2σ| (mean = 40.96%; median = 38%; *SD* = 15.27%), it should be noted that hit compounds of the **9x** and **11x** series demonstrated a greater tendency of returning inhibitory values. Remarkably, simple phenyl and *m*-F-phenyl substituents at R1 led to the best results, maintaining a pure hydrophobic nature. On the other hand, the introduction of polar groups or heterocycles at the R2 moiety level promoted the achievement of higher inhibition rates, in concordance with the electrostatics-based molecular field. At the same time, a change in the core heterocycle type into oxazole or thiazole with different connectivity and substitution at R2 with a polar aromatic substituent led to higher inhibition rates (50% or higher) compared to those of parent compounds (lower than 50%). One exception was observed for compound **13c**, which integrated a well-established *m*-F-phenyl moiety at position R1 while introducing a 3-(2-pyridonyl) group at R2, which can undergo tautomerism to generate aromatic 3-(2-hydroxy-pyridine). Such a combination remarkably outperformed all other newly designed compounds with pyridone rings.

To reinforce our findings, we performed molecular docking simulations on the investigated binding pocket specifically for the most promising hit, compound **11b**. Such simulations not only validate our initial screening results but also provide a deeper understanding of the structural features critical for binding efficacy. While reference compound SB-366791 showed a hydrogen bond with Tyr511 (within segment S3 of the channel), compound **11b** was unable to establish such interaction, thus forming only weak hydrophobic interactions with this residue (Figure 4). Nevertheless, **11b** established several hydrogen bonds, employing the 2,4-dihydroxy-phenyl moiety as a hydrogen bond acceptor for the Arg557 sidechain and the Glu570 nitrogen backbone atom and as a hydrogen bond donor towards the Gln701 sidechain carbonyl atom. Extensive hydrophobic interactions were identified, involving the opposite side of the molecule, described by the *m*-F-phenyl ring and the thiazole heterocycle and segment S4 of the channel (Phe543, Ala546, Leu547, and Thr550). Leu515 (within segment S3) was involved in the interaction with the central pyrrolidine group, which established hydrophobic contacts with Ile573 within the connection segment S4-S5 on the other side of the cleft as well. Within the same connection segment, Ala566 was responsible for the establishment of hydrophobic interactions with the 2,4-dihydroxy-phenyl ring. Lastly, the hydrophobic sidechains of Ala666 and Leu670 residues, belonging to segment S6 of the subsequent protein chain of the channel, interacted with the opposite side of the aromatic *m*-F-phenyl and thiazole rings. Overall, SB-366791 and compound **11b** returned excellent docking scores, of −8.9 and −10.0 kcal/mol, respectively.

## 3. Discussion

The main findings of the present study highlighted the possibility of designing novel human TRPV1 channel antagonists, starting from previously developed compounds. The insertion of key modifications on the original scaffolds led to the identification of a more inhibitory compound, as predicted by our 3D-QSAR model. Our model, though relying on classical molecular mechanics methods only, demonstrated good overall performance on the validation sets (*R*^2^ = 0.78). In addition, it integrated MIFs as developed within a CoMFA methodology, which is the gold-standard theory for 3D-QSAR model generation. Such a procedure interpreted each atom on the molecular grid according to an electrostatics-based force field, after alignment of the ligands to a common pharmacophore hypothesis. Since three pharmacophoric features usually represent a good balance between molecular description and adequate flexibility for small ligands, we successfully screened the training and test sets, accounting for exclusion spheres belonging to the receptor as well. The CoMFA strategy promoted the inclusion of the investigated binding site inside the 3D-QSAR model, according to the evaluated binding poses. The same rationale was followed when choosing the number of principal components to include in the PLS model, which was reinforced by the attainment of optimum accuracy when this value was equal to five. The implementation of a higher number of up to nine principal components led to an evident loss in accuracy, while selecting a minor number led to insufficient capturing of the sparsity of the data.

In the present molecular modeling approach, we evaluated an allosteric binding site to inhibit human TRPV1 channel activity. Mutagenesis on Tyr511 and Thr550 residues demonstrated that the modification of these causes reduced sensitivity to capsaicin in both rat and human TRPV1 [29,30]. In addition, hydrophobic interactions established with Leu515 and Leu547 residues were vital to observe antagonism for SB-366791 and were successfully maintained for the top-rank **11b** compound. The establishment of three extra hydrogen bonds with Arg557, Gln701, and Glu570 remarkably improved the stability of the binding pose of **11b**, leading to a more favorable docking score (−10.0 kcal/mol), if compared to reference inhibitor SB-366791 (−8.9 kcal/mol). Performing mutagenesis experiments on these residues would provide valuable insights into understanding the mechanism of action more in detail. Intriguingly, compound **11b** introduced two polar hydroxy moieties on the side aromatic ring, being one of the most polar compounds of the test set, while the substitution of the same groups with methoxy moieties led to the abrogation of activity due to steric clashes with the receptor’s sidechains. For these reasons, the original lipophilicity problem, overcome for the **7q** ligand by Qiao et al., was again encountered for the compounds of the test set, which would be unable to cross the BBB and reach the CNS. To address this issue, we predicted that compound **7q** and methoxy-substituted **11b** could be CNS-permeant, while **11b** was not (or to a very low extent). Such a finding was confirmed by two tools for the prediction of this pharmacokinetic property (lightBBB [31] and cbligand [32] online servers), after demonstrating consensus results. The former algorithm predicted **11b** as having low permeance (score of 0.379), while methoxy-substituted **11b** and **7q** compounds were predicted as being very permeant (scores of 4.613 and 4.488, respectively). The latter tool returned just a simple boolean result, classifying **11b** as non-permeant, but methoxy-substituted **11b** and **7q** compounds as permeant. Therefore, we believe that BBB delivery through the introduction of small lipophilic substituents on the two **11b** hydroxy groups, to later undisclose them by active metabolism, could be a useful strategy to overcome this issue. Nevertheless, the prodrug approach has been widely used in medicinal chemistry to fine-tune the drug candidates’ formulation and pharmacokinetics properties [33]. Taken together, our methodology demonstrated the feasibility of conducting a chemical space search by rational design, relying exclusively on open-access tools. This approach not only ensures cost-effectiveness and accessibility but also promotes transparency and reproducibility in research. The MIFs generated during our study proved to be highly reliable and outperformed the inclusion of both 2D and 3D fingerprints, highlighting their superiority in capturing the essential features required for accurate predictions.

## 4. Materials and Methods

### 4.1. Pharmacophore Modeling

TRPV1 antagonists ((S)-N-(3-isopropylphenyl)-2-(5-phenylthiazol-2-yl) pyrrolidine-1-carboxamide derivatives, previously selected after systematic in vitro and in vivo bioassays [16]) were considered as the starting point of our study (Appendix A). All the synthetic compounds were manually sketched and converted into their three-dimensional coordinates with openbabel 2.4.1 [34]. Antagonist SB-705498 was added as well as a reference compound. The obtained ligands were expanded into a multi-conformational database by generating up to 100 conformers for each compound with balloons according to the electrostatics Merck force field (MMFF94) [35,36].

Before 3D-QSAR modeling, it was vital to align the compounds on the same pharmacophore model, accounting for the best-scoring pose on the human TRPV1 receptor. The pharmacophore hypothesis was generated within the Pharmit platform [37], starting from the Cryo-EM model of the human TRPV1 channel in complex with the SB-366791 antagonist (PDB: 8GFA [18]) at 2.3 Å resolution. The SB-366791 antagonist was selected to map the pharmacophoric features (Figure 1B), along with receptor exclusion spheres set at 1.5 Å tolerance to allow for slight shape adaptability. After screening the pharmacophore with the Pharmit-compiled multi-conformational database, we selected the top-ranking conformer for each compound, matching all the selected features and showing the lowest RMSD compared to the SB-366791 ligand.

### 4.2. Three-Dimensional-QSAR Model Generation and Validation

The final 83 fitting conformers were used to generate the CoMFA-based 3D-QSAR model within the Open3DQSAR open-access tool [28]. For this modeling, the box size around the molecular ensemble and the grid step size (1.0 Å) were chosen as default parameters, and MIFs were then calculated using the MMFF94 [38]. To build the 3D-QSAR models, PLS regression was employed following the extraction of 5 top-in-rank principal components. To evaluate the performance of the model, the coefficient of *R*^2^ was computed to measure how well the model explained the variance in the data relative to the mean of the observed outcomes. Additionally, the *SD* was calculated to assess the dispersion of or variability in the predicted values around the actual values, providing insights into the consistency and precision of the model’s predictions. The formulas for *R*^2^ and *SD* are given as follows:(1)$$R^{2}=1-\frac{\sum_{i=1}^{n}{\left(y_{i}-\hat{y_{ı}}\right)}^{2}}{\sum_{i=1}^{n}{\left(y_{i}-\bar{y}\right)}^{2}},$$
(2)$$SD=\sqrt{\frac{\sum_{i=1}^{n}{\left(y_{i}-\hat{y_{ı}}\right)}^{2}}{n-1}}$$
where $y_{i}$ is the observed value, $\hat{y_{ı}}$ is the predicted value, $\bar{y}$ represents the mean of the observed values, and *n* is the number of observations.

The same process was adopted for the generation of the features of the test set of compounds, as detailed in Table 1. The compounds of the test set were designed and manually sketched by applying point modifications on the original scaffolds in the training set. A rational design approach was followed, considering the investigated binding site shape, and returned the most active ligands of the training set. A multi-conformational database was compiled as previously described and screened towards the previously built pharmacophore hypothesis.

### 4.3. Molecular Docking

The three-dimensional structure of the human TRPV1 channel (PDB: 8GFA [18]) was uploaded to the CHARMM-GUI platform [39,40,41] to model missing loops, to assign the correct protonation to sidechain residues at physiological pH, and to enclose the receptor in a membrane of 1-palmitoyl-2-oleoyl-glycero-3-phosphocholine (POPC) lipids (by placing 120 and 108 entities in the upper and lower leaflets, respectively). N- and C-termini were capped using the acetylation and carbamidomethylation modifications, respectively. In addition, the system was placed in a cubic box of water (TIP3P) at a neutralizing ionic concentration of 0.15 M NaCl. The final model, comprising 93,126 atoms, was gradually minimized to release molecular strains and steric clashes with gromacs 2022.5 [42] according to the CHARMM36m force field [43]. We utilized the steepest descent algorithm until convergence of the potential to 10 kJ/mol/nm. The energy-minimized structure served as the protein receptor during the molecular docking simulation, after its conversion into pdbqt format with openbabel as implemented inside ADFR suite 1.0 [44]. Flexibility was attributed to the Tyr511 residue of the channel during the molecular docking simulation. Compound **11b** and reference compound SB-366791 were converted into their three-dimensional coordinates in pdbqt format and used as input for the subsequent molecular docking simulation. A simulation box of 16 × 16 × 20 Å was generated by selecting SB-366791 as its center. Autodock vinaXB (AutoDock Vina, The Scripps Research Institute, 2023) was used to carry out the calculations, with exhaustiveness set to 32 from the default 8, to increase accuracy [45]. A maximum of 10 poses were generated per input structure. Interaction networks were identified with the PLIP web server [46]. Visualization and figure generation were carried out within the Pymol v2.5.2 molecular graphics interface (The PyMOL Molecular Graphics System, Version 3.0 Schrödinger, LLC).

### 4.4. BBB Permeability Prediction

The molecular structures of **11b**, methoxy-substituted **11b**, and reference **7q** compounds were provided as input to two different tools for the prediction of blood–brain barrier permeability: lightBBB [31] and cbligand [32]. For lightBBB, only the 2D SMILES representations of the molecules were used as input for the classification algorithm (either BBB-permeant or non-permeant). In contrast, for the cbligand tool, 166-bit MACCS keys were used as 2D fingerprints that were later subjected to an adaptive boosting meta-algorithm as an instance classification model (AdaBoost).

## 5. Conclusions

The present study has highlighted the potential of using previously developed compounds for the design of novel antagonists targeting the human TRPV1 channel. Based on our CoMFA-based 3D-QSAR model, we successfully identified a compound with a superior inhibition rate and favorable binding characteristics. A key aspect of our methodology was the exclusive use of open-access tools, which provided a significant advantage by enabling us to achieve excellent performance with minimal resources. This approach accelerates the research process and makes advanced computational techniques accessible to academic and low-resource settings. However, it is important to recognize the limitations of relying solely on in silico methods. While our findings offer valuable insights, they constitute only the initial phase of a more comprehensive research process. The experimental validation of the identified compounds is therefore necessary to confirm the predicted effects and ensure the reliability of our results.

In the future, dynamics studies on the flexibility of the TRPV1 channel, when treated with the compound, could be carried out to analyze the channel conformations and to observe the molecular reorganization events starting from the occupation of the allosteric binding pocket. These insights could provide a deeper understanding of the interaction mechanisms at play and guide further refinement of the antagonists. Additionally, future research will focus on optimizing BBB permeability and exploring prodrug strategies to enhance the pharmacokinetic properties of these novel compounds. In addition, the implementation of quantum-mechanical methods could lead to the improved accuracy and predictability of the compounds’ features and binding properties. By integrating these advanced techniques, we aim to refine the design and efficacy of TRPV1 antagonists, ultimately contributing to the development of more effective therapeutic agents.

## Figures and Tables

**Figure 1 ijms-25-07951-f001:**
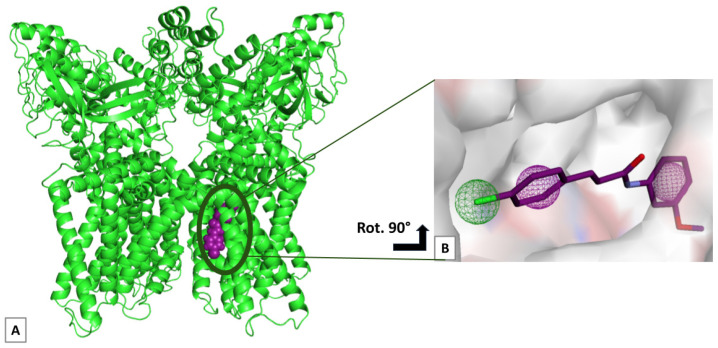
(**A**,**B**) Overview of the human TRPV1 channel in complex with inhibitor SB-366791 and the pharmacophore model. (**A**) Enlarged view of the hTRPV1 channel. The three-dimensional structure of the TRPV1 channel (green cartoon-like representation) harbors an allosteric binding site, occupied by inhibitor SB-366791 (purple spheres). (**B**) Pharmacophore model. Three pharmacophoric features are mapped on the SB-366791 bioactive pose (purple ball-and-stick representation): two aromatic features (purple mesh grid) and one hydrophobic feature (green mesh grid).

**Figure 2 ijms-25-07951-f002:**
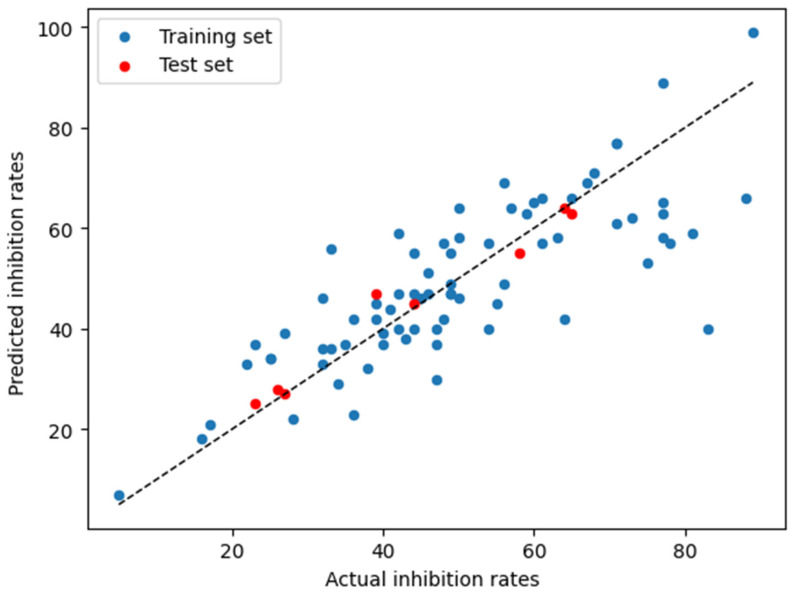
Predicted inhibition rates by PLS regression (y-axis) versus experimental inhibition rates taken from [16] (x-axis) for the training set (blue dots) and validation set (red dots).

**Figure 3 ijms-25-07951-f003:**
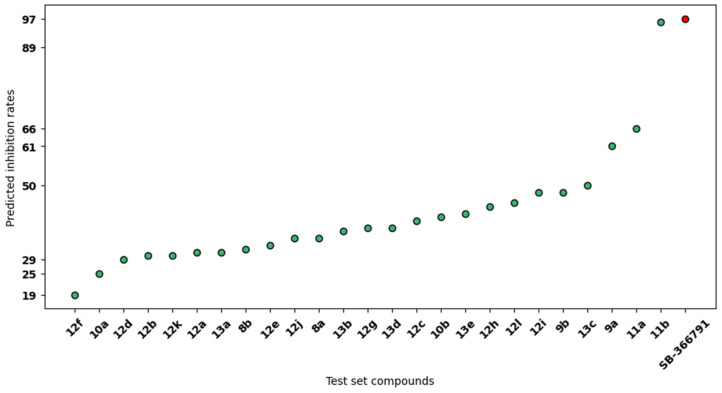
Predicted inhibition rates by PLS regression for the test set. For each of the newly designed compounds (x-axis); their predicted inhibition rates by PLS regression are reported (y-axis) according to the 3D-QSAR model. The red point represents the reference compound SB-366791, allowing for a direct comparison of its inhibition rates against those of the newly designed compounds.

**Figure 4 ijms-25-07951-f004:**
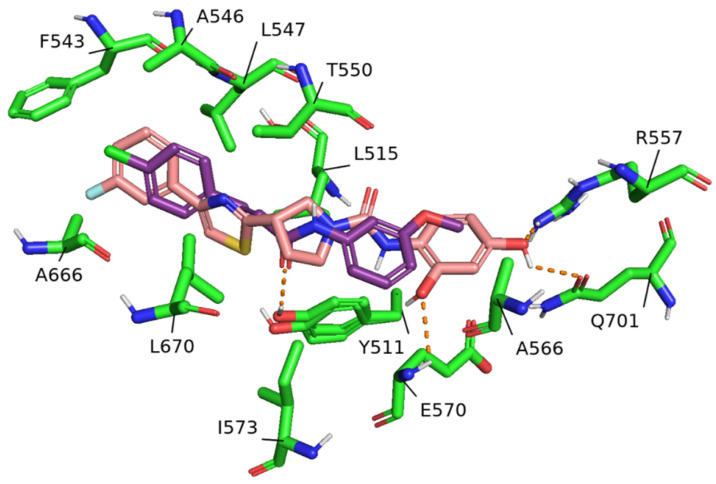
Comparative binding poses of SB-366791 and **11b** after molecular docking. The interaction network of the two compounds within the hTRPV1 channel is shown in the picture. Interacting residues are depicted in green ball-and-stick representation. SB-366791 (purple ball-and-stick representation) forms a hydrogen bond with Tyr511 (orange dashed lines). **11b** (pink ball-and-stick representation) establishes three hydrogen bonds with Arg557, Gln701, and Glu570 (orange dashed lines). Hydrophobic interactions are not explicitly marked for the sake of clarity.

**Table 1 ijms-25-07951-t001:** Test set of compounds **8a**,**b**, **9a**,**b**, **10a**,**b**, **11a**,**b**, **12a**–**l**, **13a**–**e**. The numeric prefix in the names of the series of compounds refers to diverse initially modified scaffolds.

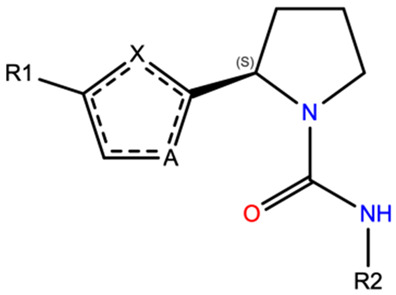				
Compounds	R1	R2	A	X
**8a**	phenyl	4-pyrimidinyl	S	N
**8b**	phenyl	4-(2-Cl-pyrimidinyl)	S	N
**9a**	phenyl	4-pyrimidinyl	N	S
**9b**	phenyl	phenyl	N	O
**10a**	*p*-F-phenyl	4-pyridinyl	S	N
**10b**	*p*-F-phenyl	2-thiophenyl	S	N
**11a**	*m*-F-phenyl	5-imidazolyl	S	N
**11b**	*m*-F-phenyl	2,4-dihydroxy-phenyl	S	N
**12a**	*m*-Me-phenyl	*m*-iPr-phenyl	S	N
**12b**	3-F-4-Me-phenyl	*m*-iPr-phenyl	S	N
**12c**	phenyl	5-(3-iPr-pyridinyl)	S	N
**12d**	phenyl	4-(2-iPr-pyridinyl)	S	N
**12e**	3-(1,2,5-triMe-pyrrolyl	*m*-iPr-phenyl	S	N
**12f**	3-(2,5-diMe-thiophenyl)	*m*-iPr-phenyl	S	N
**12g**	3-thiophenyl	*m*-iPr-phenyl	S	N
**12h**	2,2′-diMe-phenyl	*m*-iPr-phenyl	S	N
**12i**	*m*-F-phenyl	2-OMe-4-hydroxy-phenyl	S	N
**12j**	*m*-F-phenyl	1-naphtalyl	S	N
**12k**	*m*-F-phenyl	2,3-diMe-phenyl	S	N
**12l**	*m*-F-phenyl	2,3-diMe-4-hydroxy-phenyl	S	N
**13a**	phenyl	4-(2-pyridonyl)	S	N
**13b**	phenyl	3-(2-pyridonyl)	S	N
**13c**	*m*-F-phenyl	3-(2-pyridonyl)	S	N
**13d**	4-(2-pyridonyl)	*m*-iPr-phenyl	S	N
**13e**	5-(2-pyridonyl)	*m*-iPr-phenyl	S	N

## Data Availability

The data presented in this study are available on request from the corresponding author.

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
