# Peer review of "Three-Dimensional Quantitative Structure–Activity Relationship Study of Transient Receptor Potential Vanilloid 1 Channel Antagonists Reveals Potential for Drug Design Purposes"

_ijms, 2024, doi:10.3390/ijms25147951_

Round 1

Reviewer 1 Report

Comments and Suggestions for Authors

The study by Gianibbi et al. presents a commendable effort in utilizing a virtual screening approach to identify potential TRPV1 inhibitors. The use of the Cryo-EM 3D structure of a nanomolar antagonist complexed with the human TRPV1 channel to build a pharmacophore model is a sophisticated and innovative strategy. The identification of compound 11b, which is suggested to have a more potent effect on TRPV1 than the well-known inhibitor SB-366791, is particularly intriguing and holds significant promise for clinical applications, especially in the context of chronic pain management.

However, it is crucial to recognize the limitations of relying solely on in silico methods. While the computational results provide a strong foundation, they cannot substitute for experimental validation. To draw robust and reliable conclusions, it is imperative that the authors perform functional validation of their new drug candidate. Specifically, electrophysiological experiments are absolutely essential to confirm the predicted potency and efficacy of compound 11b in modulating TRPV1 activity. Without this critical validation step, the conclusions drawn from the simulation results remain speculative.

Moreover, in Figure 3, it is recommended that the authors include SB-366791 as a reference compound. This will provide a valuable benchmark and facilitate a direct comparison between the newly identified compound 11b and the established inhibitor. 

In summary, while the computational approach and findings are noteworthy, experimental validation through electrophysiological studies is indispensable. 

Author Response

The study by Gianibbi et al. presents a commendable effort in utilizing a virtual screening approach to identify potential TRPV1 inhibitors. The use of the Cryo-EM 3D structure of a nanomolar antagonist complexed with the human TRPV1 channel to build a pharmacophore model is a sophisticated and innovative strategy. The identification of compound 11b, which is suggested to have a more potent effect on TRPV1 than the well-known inhibitor SB-366791, is particularly intriguing and holds significant promise for clinical applications, especially in the context of chronic pain management.

However, it is crucial to recognize the limitations of relying solely on in silico methods. While the computational results provide a strong foundation, they cannot substitute for experimental validation. To draw robust and reliable conclusions, it is imperative that the authors perform functional validation of their new drug candidate. Specifically, electrophysiological experiments are absolutely essential to confirm the predicted potency and efficacy of compound 11b in modulating TRPV1 activity. Without this critical validation step, the conclusions drawn from the simulation results remain speculative.

Thank you for your insightful comment. As suggested, we will undertake this critical validation step soon. However, the primary aim of our current project is to emphasize that an initial in-silico exploration is crucial for guiding future studies, which may delve deeper into the functional validation of the identified compounds.

In summary, based on the proposed suggestions emphasizing the importance of experimentally validating the predicted potency and efficacy of compound 11b, we have revised our conclusions. We now underscore that while our findings offer valuable insights, they represent only the initial phase of a more comprehensive research process.

Moreover, in Figure 3, it is recommended that the authors include SB-366791 as a reference compound. This will provide a valuable benchmark and facilitate a direct comparison between the newly identified compound 11b and the established inhibitor. 

Thank you for your observation. We have included the reference compounds in the image.

In summary, while the computational approach and findings are noteworthy, experimental validation through electrophysiological studies is indispensable. 

Reviewer 2 Report

Comments and Suggestions for Authors

The manuscript on "3D-QSAR study of TRPV1 channel antagonists reveals potential for drug design purposes" by Gianbibbi et al. is a very interesting small case study showing the potential of open-source and open-access software. I really welcome such clear demonstration that 3D-QSAR studies could be handled easily without reference to the copyrighted, in many cases obsolete or no-longer-distributed, software.

The manuscript definitely should be published in the International Journal of Molecular Sciences, with one important minor addition. The manuscript is solely theoretical and it uses (as far as I understand) previously published compounds. It is therefore necessary to put the information on the source of activities very close to the data itself, e.g. in the caption of Figure 2 ("... experimental inhibition rates taken from Ref. [...]"). It would be also optimal to add these data to the Supplementary material.

Otherwise, I find the manuscript clearly written and, as noted above, I do recommend its publication. The iThenticate report indicates high degree of independence of the manuscript from others in the field, and I do not have any reservations to methodology or the research itself.

Author Response

The manuscript on "3D-QSAR study of TRPV1 channel antagonists reveals potential for drug design purposes" by Gianbibbi et al. is a very interesting small case study showing the potential of open-source and open-access software. I really welcome such clear demonstration that 3D-QSAR studies could be handled easily without reference to the copyrighted, in many cases obsolete or no-longer-distributed, software.

The manuscript definitely should be published in the International Journal of Molecular Sciences, with one important minor addition. The manuscript is solely theoretical and it uses (as far as I understand) previously published compounds. It is therefore necessary to put the information on the source of activities very close to the data itself, e.g. in the caption of Figure 2 ("... experimental inhibition rates taken from Ref. [...]"). It would be also optimal to add these data to the Supplementary material.

Thank you for your observation. We have modified the caption of Figure 2 accordingly and added the experimental data to the Supplementary material.

Otherwise, I find the manuscript clearly written and, as noted above, I do recommend its publication. The iThenticate report indicates high degree of independence of the manuscript from others in the field, and I do not have any reservations to methodology or the research itself.

Reviewer 3 Report

Comments and Suggestions for Authors

The manuscript “3D-QSAR study of TRPV1 channel antagonists reveals potential for drug design purposes” by Beatrice Gianibbi et al. virtually screened potential antagonists of the human TRPV1 channel. Pharmacophore modeling of previously developed compounds identified three pharmacophoric features, which were then implemented to establish a 3D-QSAR model. The validated model led to the design of more promising new compounds. The study has laid the foundation for future in-depth in-silico and wet-lab drug development. I think this concise and well-structured manuscript is worthy of publication in Int. J. Mol. Sci., with only minor concerns to address. Further review is not needed.

1. I found references were messy. E.g., :

1) Page 9, line 331: “uploaded to the CHARMM-GUI platform [39-41]”. Refs. 39–41 have nothing to do with the CHARMM-GUI. Refs. 44–46 are relevant.

2) Page 10, line 339: “gromacs 2022.5 according to CHARMM36m force field [42, 43]”. Refs. 42–43 are irrelevant. Ref. 39 refers to the CHARMM36m force field. Ref. 40 refers to the GROMACS 2022.5 software.

Please thoroughly check references to ensure accurate citation.

Author Response

The manuscript “3D-QSAR study of TRPV1 channel antagonists reveals potential for drug design purposes” by Beatrice Gianibbi et al. virtually screened potential antagonists of the human TRPV1 channel. Pharmacophore modeling of previously developed compounds identified three pharmacophoric features, which were then implemented to establish a 3D-QSAR model. The validated model led to the design of more promising new compounds. The study has laid the foundation for future in-depth in-silico and wet-lab drug development. I think this concise and well-structured manuscript is worthy of publication in Int. J. Mol. Sci., with only minor concerns to address. Further review is not needed. I found references were messy. E.g., :

1) Page 9, line 331: “uploaded to the CHARMM-GUI platform [39-41]”. Refs. 39–41 have nothing to do with the CHARMM-GUI. Refs. 44–46 are relevant.

2) Page 10, line 339: “gromacs 2022.5 according to CHARMM36m force field [42, 43]”. Refs. 42–43 are irrelevant. Ref. 39 refers to the CHARMM36m force field. Ref. 40 refers to the GROMACS 2022.5 software.

Please thoroughly check references to ensure accurate citation.

Thank you for your comment. We have thoroughly rechecked the references to ensure accurate citation and clarity.

Round 2

Reviewer 1 Report

Comments and Suggestions for Authors

Agree to publish